# Y98 Mutation Leads to the Loss of RsfS Anti-Association Activity in *Staphylococcus aureus*

**DOI:** 10.3390/ijms231810931

**Published:** 2022-09-18

**Authors:** Bulat Fatkhullin, Alexander Golubev, Natalia Garaeva, Shamil Validov, Azat Gabdulkhakov, Marat Yusupov

**Affiliations:** 1Institute of Protein Research, Russian Academy of Science, 142290 Pushchino, Russia; 2Department of Integrated Structural Biology, Institute of Genetics and Molecular and Cellular Biology, INSERM, U964, CNRS, UMR7104, University of Strasbourg, 67400 Illkirch Graffenstaden, France; 3Laboratory for Structural Analysis of Biomacromolecules, Kazan Scientific Center of Russian Academy of Sciences, 420111 Kazan, Russia; 4Laboratory of Structural Biology, Institute of Fundamental Medicine and Biology, Kazan Federal University, 420021 Kazan, Russia

**Keywords:** ribosome, maturation factor, hibernation factor, RsfS, *Staphylococcus aureus*

## Abstract

Ribosomal silencing factor S (RsfS) is a conserved protein that plays a role in the mechanisms of ribosome shutdown and cell survival during starvation. Recent studies demonstrated the involvement of RsfS in the biogenesis of the large ribosomal subunit. RsfS binds to the uL14 ribosomal protein on the large ribosomal subunit and prevents its association with the small subunit. Here, we estimated the contribution of RsfS amino acid side chains at the interface between RsfS and uL14 to RsfS anti-association function in *Staphylococcus aureus* through in vitro experiments: centrifugation in sucrose gradient profiles and an *S. aureus* cell-free system assay. The detected critical Y98 amino acid on the RsfS surface might become a new potential target for pharmacological drug development and treatment of *S. aureus* infections.

## 1. Introduction

The ribosome is a ribonucleoprotein complex responsible for protein synthesis in all living organisms. The bacterial ribosome is composed of the small (30S) subunit, which contains 16S rRNA with about 20 proteins, and the large (50S) ribosomal subunit, which consists of the 23S and 5S rRNAs with more than 30 proteins [1]. The process of protein translation is energy-consuming and involves up to 40% of the energy turnover of the cell; therefore, translation is strictly regulated in different stages [2]: by biogenesis factors in the ribosomal subunit assembly stage [2,3], in the translation process [4,5,6], and by different stress factors under stress conditions [7,8,9].

At present, knowledge of the bacterial stress factor RsfS (Ribosomal silencing factor S) is very limited, and only a few research papers have been published on this topic [10,11,12,13,14]. The first model of the RsfS–50S ribosomal subunit complex was suggested based on mutagenesis experiments of the *Escherichia coli* ribosomal protein uL14, which is the main partner of RsfS in the complex with the large subunit [10,11]. *E. coli* rsfs knocked-out cells demonstrated decreased viability in a stationary growing phase and during the transition from rich to poor media [11]. A low-resolution (9 Å) cryo-EM structure of the 50S–RsfS complex from *Mycobacterium tuberculosis* showed an RsfS spatial orientation on the 50S subunit surface [12]. An X-ray structure of the RsfS–uL14 heterodimer from *Staphylococcus aureus* revealed molecular details of the interaction interface (Figure 1a,c) [13]. A high-resolution (3.2 Å) cryo-EM structure of the 50S–RsfS complex from *S. aureus* showed additional potential contacts of RsfS with the ribosomal protein bL19 and helix H95 of 23S ribosomal RNA (rRNA) (Figure 1b,d) [13]. The results of a sucrose gradient analysis demonstrated the anti-association activity of RsfS under semi-dissociation conditions [13]. According to the latest high-resolution (2.4 Å) cryo-EM structure of RsfS from *E. coli*, RsfS also binds with an unmatured large ribosomal subunit, and it interacts with the ribosomal biogenesis factor ObgE and stimulates its GTPase activity by up to 45% [14]. The bacterial factor RsfS has a similar function to its mitochondrial orthologue MALSU1 (Mitochondrial Assembly of ribosomal Large Subunit 1). MALSU1 was found in several cryo-EM structures of human mitochondrial large ribosomal subunit maturation steps. MALSU1 participates in the prevention of premature subunit association in the initiation of translation, recycling and ribosome biogenesis [15,16,17,18,19,20]. In the complex with the large ribosomal subunit, MALSU1 is in contact with the ribosomal proteins uL14 and bL19m and the sarcin–ricin loop (H95) of mitochondrial rRNA [15]. It also has a connection with mitochondrial homologues of bacterial ObgE, EF-Tu and RbgA [18,19,20].

Thus, the bacterial factor RsfS controls the translation level in a stationary growing phase or during starvation, and according to the latest research, it could also act as a ribosome biogenesis factor. RsfS is an essential bacterial protein, and since the full scope of RsfS functions is still not clear, more studies of Rsfs are in demand. In the present article, we identified the positions of amino acid residues of *S. aureus* RsfS and bonds they form that are important for the anti-association activity in *S. aureus*. We showed the influence of RsfS amino acid side chains through sucrose density gradient ultracentrifugation and an *S. aureus* cell-free protein translation assay. The position found on the RsfS surface could become a target for antimicrobial drugs binding the region in order to break the RsfS function and weaken the pathogenic properties of *S. aureus*.

## 2. Results

### 2.1. Sucrose Gradient Analysis

We chose four amino acids on the RsfS surface that form contacts with uL14 (E70, D81, W77, and Y98) and that are supposedly important for the formation of the RsfS–50S complex (Figure 1c,d). To verify our assumption, we performed single substitutions of the chosen amino acid residues for Ala. Only two mutations, E70A and Y98, provided soluble recombinant RsfS proteins. We added the 2X molar excess of purified RsfS proteins to vacant 70S ribosomes because such a concentration of proteins is optimal for sucrose gradient profile analysis [13]. Since RsfS has an anti-association effect on ribosomal subunits, and the association of bacterial ribosomal subunits is most sensitive to the concentration of magnesium ions (Mg^2+^) [21], we decided to titrate Mg^2+^ at a constant concentration of monovalent salts to test the effect of RsfS mutations on the relative proportions of the free 50S subunits and 70S ribosomes (Figure 2a).

The comparison of sucrose profiles showed small differences between the control profiles for the vacant 70S ribosomes and the RsfS Y98A profiles at 3.5 mM to 5 mM Mg^2+^. The overwhelming majority of 70S ribosomes dissociated into 50S and 30S subunits at lower concentrations (3.0 mM to 3.5 mM) of Mg^2+^ in cases with the addition of the wild-type RsfS and RsfS E70A. We continued to observe peaks for 50S and 30S subunits under all conditions in cases with the addition of the wild-type RsfS and RsfS E70A, but for the wild type, the subunit peaks were more pronounced. The relative amount of dissociated ribosomal subunits demonstrated dissociation of half of the 70S ribosomes at 5 mM Mg^2+^ for the wild-type RsfS, at around 4.4 mM Mg^2+^ for RsfS E70A and at 3.4 mM Mg^2+^ for RsfS Y98A (Figure 2b). These results demonstrate a decrease in RsfS mutants’ ability to shift the association equilibrium in the association condition (high concentration of Mg^2+^) toward subunit dissociation and indirectly indicate a decrease in the efficiency of RsfS binding to 50S ribosome subunits. Both mutants lose their anti-association effect compared to the wild type in the same conditions, but the Y98 mutation is more critical for RsfS binding than E70. The most representative concentration of Mg^2+^ was 4 mM. At this point, the wild-type RsfS kept its activity near 100%, and the E70A mutant lost around 20%, whereas the Y98A mutant lost about 80% of its anti-association activity.

### 2.2. Cell-Free System Analysis

For the evaluation of the RsfS binding efficiency with absent amino acid residue side chains, we decided to check the influence of RsfS and its mutants on the translation process. We developed an *S. aureus* cell-free in vitro translation system and performed kinetics experiments with sfGFP as a fluorescent reporter. Since the cell-free in vitro transcription–translation reaction is a complex multi-component system, RsfS could lose binding efficiency due to reasons beyond our control. Therefore, we decided to add the 10X molar excess of purified RsfS proteins to the calculated number of ribosomes in the cell lysate to prevent incorrect results due to uncontrolled RsfS efficiency loss. The same RsfS:ribosome ratio did not lead to a change in RsfS activity in previous research [13]. We conducted kinetics experiments at three different concentrations of Mg^2+^: 6 mM, 8 mM and 10 mM (Figure 3a).

The addition of RsfS or its mutants slowed down GFP translation (decrease in the slope of the fluorescence curves) by trapping the dissociated 50S subunits and preventing their reassociation with the small subunit. The highest efficiency corresponded to the wild-type RsfS, followed by RsfS E70A, and the lowest efficiency corresponded to RsfS Y98A. The increase in the Mg^2+^ concentration led to a decrease in the efficiency of translation inhibition by RsfS, and to convergence of the curves, which overlapped at 10 mM Mg^2+^. These results are consistent with the endpoint fluorescence measurements (Figure 3b). The endpoint describes the portion of irreversibly inactivated ribosomes (50S subunits) by RsfS and its mutants, or the efficiency of translation inhibition, which also indirectly indicates the efficiency of RsfS binding to large subunits. The efficiency of inhibition was less than 20% for RsfS E70A and less than 50% for RsfS Y98A compared to the wild-type RsfS at 6 mM of Mg^2+^; in the case with 8 mM Mg^2+^, these values were ~15% and ~25%, respectively. All RsfS proteins lost their activity at a high (10 mM) concentration of Mg^2+^.

## 3. Discussion

The aim of our research was to expand the knowledge about ribosome silencing and the viability of *S. aureus* cells under stress conditions. The main participants of the ribosome silencing process are the RsfS and uL14 proteins, which form a heterodimer [13]. The binding of a small drug or a peptide antibiotic to the interface of the heterodimer or its border could prevent the heterodimer association. We propose that RsfS could be a perspective specific drug target, in contrast to ul14, which is a highly conserved protein (Supplementary Figure 3 in [13]). We analyzed the X-ray structure of the heterodimer from *S. aureus* [13] and assumed that E70, D81, W77 and Y98 of RsfS could be the most important amino acids for the stability of the heterodimer. Only E70 and D81 formed two potential H-bonds with uL14; W77 and Y98 interacted with uL14 through massive hydrophobic side chains; and Y98 formed an H-bond with P93 of uL14 through the contact with a water molecule (Figure 1c). E70 saved the same amount of potential H-bonds in the cryo-EM model; the Y98 side chain formed an H-bond with P93 directly; W77 had the same conformation as in the X-ray structure; and only D81 lost both potential H-bonds (Figure 1d). We performed a single replacement of the chosen amino acids by alanine to deprive interactions that formed through this amino acid side chain. Other contacts between RsfS and the 50S subunit were not changed. Having carried out all the above, we could estimate the contribution of each chosen side chain.

In our studies, the expression of two mutants of RsfS (W77 and D81) did not yield soluble recombinant proteins. The researchers that studied RsfS from *M. tuberculosis* faced a similar problem: only 1 (E74A) of 14 single replacement mutants was soluble [12]. Thus, we could assume that the RsfS protein has low tolerance to the replacement of amino acids situated on the interface between RsfS and uL14. This could be a consequence of the incorrect folding of RsfS mutants. We checked the soluble RsfS and its mutants (E70A and Y98A) through an additional step of gel filtration before the experiments to prevent incorrect interpretation of the results due to aggregation (Appendix A). According to the obtained sucrose gradient profiles (Figure 2a) and gel filtration profiles (Appendix A), the RsfS wild type and the two mutants (E70A and Y98A) were stable and had anti-association activity. This means the loss of mutants’ activity is not connected to protein aggregation and depends on amino acid replacement. The relative amount of dissociated ribosomal subunits demonstrates the importance of conditions under which RsfS activity is compared. A comparison under single conditions of Mg^2+^ could be incorrect and might not show a real influence of the mutations. We suggest using a condition screening and evaluation of the change in the most sensitive point for each experiment. In our case, it was 4 mM Mg^2+^ for the sucrose gradients and 6 mM Mg^2+^ (370 mM K+) for the *S. aureus* cell-free system. According to our proposal, the RsfS E70A mutant lost about 20% of its anti-association activity and demonstrated 70% efficiency of translation inhibition on average; the RsfS Y98A mutant lost about 80% of its anti-association activity and demonstrated approximately 35% efficiency of translation inhibition; and the RsfS wild type showed 100% anti-association activity and approximately 80% efficiency of translation inhibition. Thus, Y98A is the most critical mutation for RsfS activity.

The *E. coli* RsfS wild type showed translation inhibition efficiency of about 80% under the 4.5 mM Mg^2+^ condition [11]; for *M. tuberculosis*, the translation inhibition efficiency was 84% for the RsfS wild type and 20% for RsfS E74A under the 2.4 mM Mg^2+^ condition [12]. The E74 amino acid side chain of *M. tuberculosis* RsfS that is significant for the protein activity corresponds to E70 of *S. aureus* RsfS. Unfortunately, we cannot perform a comparative analysis of their results concerning the loss of RsfS mutant efficiency, as there is no information on the dependence of RsfS efficiency on Mg^2+^ conditions. We also cannot analyze the structural reasons for such different activities due to the lack of a high-resolution structure of Rsfs in complex with uL14 or with the 50S subunit from *M. tuberculosis*, but we can definitely state that E74 is more important for *M. tuberculosis* than E70 is for *S. aureus*.

In the PDB bank, only two high-resolution structures of RsfS in complex with the 50S subunit are available: the high-resolution (2.4 Å) cryo-EM complex from *E. coli* (PDB: 7BL4), and the high-resolution (3.2 Å) cryo-EM complex from *S. aureus* (PDB: 6SJ6). The interface areas of the RsfS–uL14 heterodimer are close in both complexes: 888 Å^2^ in *E. coli*, and 819 Å^2^ in *S. aureus*. The number of potential contacting interface amino acids is also the same: seven H-bonds and eight amino acids in hydrophobic interactions (Appendix A) (Figure 4a,b). Therefore, the loss of one identical side group could equally affect the anti-association activity in *E. coli* and *S. aureus*. 

According to previous mutagenesis studies of uL14 interface amino acid residues to Ala in *E. coli*, the K114A and T97A amino acids are the most critical, R98A is important and S117A does not influence uL14 and RsfS binding [11] (Figure 4a). These amino acids correspond to K113, T96, R97 and S116 in uL14 of *S. aureus*, respectively (Figure 4b). The analysis of inner heterodimer contacts from *E. coli* showed the absence of any bonds between RsfS and the side chain of S117 of uL14 (Appendix A), as with S116 of uL14 in *S. aureus* [13]. The location and surroundings of the conservative K114 are the same as for K113 of uL14 from *S. aureus*: they both form one H-bond through a side chain (Appendix A), and we assume that the side chain of K113 could also be critical for heterodimer formation. An interesting moment occurs with the replacement of the conservative amino acids T97A and R98A in uL14 from *E. coli* [11]. The important R98 is located at the periphery of the heterodimer interface and forms one H-bond with the main-chain oxygen of RsfS (as well as R97 of uL14 from *S. aureus*), while the critical T97 of uL14 from *E. coli* does not form an H-bond or hydrophobic contacts with RsfS amino acids (Figure 4a). T96 of uL14 from *S. aureus* also does not form any contact with RsfS, but it is located near the critical Y98 (discovered in this work) (Figure 4b). The accessibility of solvent molecules to the hydrophobic center could be the reason for the importance of R98 and K114 of uL14 from *E. coli* in heterodimer formation. The CASTp service [22] calculation confirmed this through changes in the intermolecular solvent-accessible surfaces around the hydrophobic center (W77) for heterodimers with single substitutions. The solvent molecules’ accessibility increases in the cases of K114A and R98A. T97A allows solvent molecules’ access to Y98, but not to the hydrophobic center. S117A does not lead to any surface changes in the heterodimer from *E. coli* (Appendix B
Figure A1). The substitution of E70A does not allow the access of water molecules to W77 in *S. aureus* because K113 blocks it (Figure A2a). The Y98A substitution leads to a vast increase in the solvent-accessible surface in the RsfS–uL14 heterodimer (especially around W77), which most likely leads to its destabilization (Figure A2b).

In summary, the heterodimers are destabilized in cases of deep solvent penetration into hydrophobic surfaces (R98A and K114A substitutions in uL14 from *E. coli* and Y98A RsfS from *S. aureus*). Since T97A is critical for heterodimer formation in *E. coli*, and a wide increase in the solvent-accessible surface is absent, we therefore assume that T96 (T97) of uL14 coordinates and stabilizes the Y98 side chain of RsfS in the right position, which blocks a large hydrophobic surface (Appendix B
Figure A1 and Figure A2) in the center of RsfS from the solvent. Despite the high conservation of this region, the *S. aureus* RsfS Y98 amino acid is surrounded by the non-conserved amino acids R76, E94 and Y97 (Supplementary Figure 3 in [13]); this site could become the binding pocket for specific drugs (Figure 4c) that change the conformation of Y98 to prevent the RsfS association with uL14 and the 50S subunit.

## 4. Materials and Methods

### 4.1. Expression and Purification of S. aureus RsfS and Its Mutants

RsfS with uL14 of *S. aureus* was cloned into a modified pACYCDuet-1 plasmid with a six-histidine tag fused at the N-terminus of uL14. Quick-change mutagenesis was used to obtain four RsfS mutants with a single replacement: E70A, D81A, W77A and Y98A. The wild-type RsfS and its mutants were co-expressed with uL14 in *E. coli* BL21star (DE3) cells in LB broth. Protein expression was induced with 0.5 mM IPTG when an OD_600_ of 0.6 was reached. The cells were harvested after 4 h following induction at 37 °C. Upon cell disruption through sonication in Buffer A (50 mM Tris-HCl, 500 mM NH_4_Cl, pH 8.0) supplemented with phenylmethylsulphonyl fluoride (PMSF), the cell lysate was cleared through 1 h ultracentrifugation at 215,000× *g*. The supernatant was purified from uL14 through Ni-NTA chromatography. The flow-through was diluted 10 times using Buffer B (50 mM Tris-HCl, pH 8.0) and purified through affinity chromatography (heparin sepharose). The flow-through from heparin sepharose was further purified through ion-exchange chromatography (Q-sepharose), and proteins were eluted in a 0.1 M to 0.7 M NaCl step gradient in Buffer B. As a final purification step, the proteins were applied to size-exclusion chromatography using a Superdex 200 Increase 10/300 GL column equilibrated in Buffer C (10 mM Tris-Acetate pH 8.2, 60 mM KOAc, 14 mM MgAc_2_) for cell-free experiments and in Buffer D (0.2 M NaCl, 50 mM Tris-HCl, pH 8.0) for sucrose gradient analysis (Appendix A). The collected fractions were snap-frozen in liquid nitrogen and stored at −80 °C. The purity of samples was verified by SDS-PAGE and an additional step of size-exclusion chromatography (Appendix A).

### 4.2. Sucrose Gradient Analysis

Vacant *S. aureus* 70S ribosomes were obtained using a sucrose cushion containing 500 mM KCl, sucrose gradients of 0–30% and precipitation with PEG 20,000 following a previously published protocol [13]. For magnesium titration experiments, 70S ribosomes were dialyzed in Buffer G (10 mM Hepes-KOH pH 7.5, 50 mM KCl, 10 mM NH_4_Cl) containing the respective Mg^2+^ concentration for at least 3 h. To evaluate RsfS dissociation activity, 70S ribosomes were incubated with the 2X molar excess of the purified RsfS protein or its mutants for 1 h at 37 °C and loaded onto sucrose gradients prepared in Buffer G. All sucrose gradients performed in this study were prepared with a linear 0–30% gradient of sucrose in Buffer G supplemented with different concentrations of Mg-acetate (3 mM, 3.5 mM, 4 mM, 4.5 mM and 5 mM). All centrifugations were performed using a Beckman SW41 rotor running at 65,000× *g* for 14 h at 4 °C. Fractions of 0.4 mL were collected and measured using a Nanodrop 2000 (Thermo Scientific™) at a 260 nm wavelength. Two different preparations of the 70S sample were used in this study.

### 4.3. S. aureus Cell-Free System Preparation

To begin the preparation of cells, we plated the *S. aureus* strain 4220 from a glycerol stock on LBA plates overnight. The next day, we prepared a culture in LB and grew it at 180 rpm and 37 °C in a shaker overnight. Then, the overnight culture was taken and inoculated into new flasks with LB and grown at 180 rpm and 37 °C until the mid-log phase (1–1.2 OD_600_). We cooled the flasks down in an ice-NaCl bath for approximately 15 min to stop the growth and then spun them at 4000 rpm and 4 °C for 15 min. Cells were then washed 3 times in Buffer A (10 mM Tris-Acetate pH 8.2, 14 mM MgOAc, 60 mM KCl) and spun at 4000 rpm and 4 °C for 15 min. The cells were weighed and stored at −80 °C.

To prepare the protein extract, we disrupted the cells through enzymatic lysis. Cells were thawed on ice and resuspended in Buffer A with added Protein Inhibitor Cocktail (Roche) in a 50 mL Falcon tube. Then, we added lysostaphin—1 mg of lysostaphin to 1 g of cells—and adjusted the volume with Buffer A to 15 mL. The sample was incubated in a 37 °C water bath for 50 min with gentle mixing. Then, mild sonication on ice was used to remove the viscosity of the lysate, with 20 pulses for 2 s followed by a 3 s pause at 20% amplitude (Bioblock scientific, Vibra Cell). The ultrasound-treated sample was then spun at 30,000× *g* and 4 °C for 30 min, and the upper nine-tenth of the supernatant was taken for the next centrifugation; the supernatant was spun again at 30,000× *g* and 4 °C for 30 min, and the upper nine-tenth was taken. Then, the extract was processed using a run-off reaction. The sample was placed in a 15 mL Falcon tube covered with tin foil at 37 °C for 80 min with slight shaking. Then, the extract was dialyzed against Buffer B (10 mM Tris-Acetate pH 8.2, 14 mM MgOAc, 60 mM KOAc, 0.5 mM DTT) in an RC 12–14 kDa dialysis bag at 4 °C. The buffer was changed after 2 h, and then the dialysis was carried out overnight. Finally, the extract was spun at 10,000 rpm and 4 °C for 10 min to remove any residual particles prior to freezing, aliquoted and snap-frozen in liquid nitrogen. The concentration of extracts was measured by the Bradford assay [23]. 

On the measurement day, we mixed the cell-free reaction containing: 35% of the 10–20 mg/mL protein extract, 9 mM Mg-acetate, 230 mM KOAc (final concentration of K+ equaled 370 mM), 2 mM DTT, 0.5 mM of each amino acid and 1 mM of additional RCWMDE amino acids, 100 mM HEPES pH 8, 1.2 mM ATP, 0.8 mM CTP, UTP and GTP, 0.5 mg/mL *E. coli* tRNA, 20 mM Acetyl Phosphate, 0.1 mg/mL folinic acid, 1.5 mM spermidine, 0.1 mM EDTA-(Na), 2% PEG-8000, 0.04 mg/mL Pyruvate kinase, 20 mM Phospho(enol)pyruvic acid, 0.05% Sodium azide, RiboLock RNase inhibitor 0.3 U/mkl, 1x Complete^®^ protease inhibitor, 5 U/mkl T7 polymerase and pJL1-sfGFP plasmid (Addgene #102634) (BioBits Bright: A fluorescent synthetic biology education kit) 40 ng/mkl. The plasmid was purified with Qiagen plasmid MaxiKit without RNAse treatment and then additionally with the CaCl_2_/PEG precipitation technique (sequential CaCl_2_, polyethylene glycol precipitation for RNase-free plasmid DNA isolation). The final concentrations of magnesium and potassium are an object of optimization for each batch of an extract; in general, the optimum conditions are 14 mM for Mg^2+^ and 370 mM for K^+^. After thawing, the extract was clarified by centrifugation for 10 min at 14,000 rpm and 4 °C prior to the addition to the mix. The reaction mix was held on ice during the setup. After the master mix was prepared, it was aliquoted into different tubes to change the reaction conditions for measurements.

### 4.4. Kinetic Experiments of GFP Fluorescence in a Cell-Free System

For the kinetics experiments, cell-free mixtures were prepared with different concentrations of Mg-acetate (6 mM, 8 mM and 10 mM). Cell-free mixtures were mixed with the 10X molar excess of the purified RsfS protein or its mutants and poured into wells with the sfGFP plasmid added, excluding the negative control. All samples were replicated three times and kept at 30 °C during the experiment. GFP fluorescence was measured every 30 min for 12 h for the negative control (cell-free mixture without the sfGFP plasmid), the positive control (cell-free mixture with the sfGFP plasmid added), RsfS_WT (cell-free mixture with the sfGFP plasmid and wild-type RsfS protein added), RsfS_E70A (cell-free mixture with the sfGFP plasmid and RsfS E70A mutant protein added) and RsfS_Y98A (cell-free mixture with the sfGFP plasmid and RsfS Y98A mutant protein added) simultaneously. GFP fluorescence for the endpoint experiments was measured after 2 h from the end of the kinetics experiments. All measurements of GFP fluorescence were conducted in 384-well Greiner black plates on a PHERAstar Plus Microplate reader. Wavelengths for the sfGFP measurement were 485 nm (Ex) and 510 nm (Em) with a 5 nm bandwidth.

### 4.5. Calculations and Data Analysis

The sucrose gradient profiles were normalized by their area for the evaluation of the relative amount of dissociated ribosomal subunits (50S + 30S) to all subunits (70S + 50S + 30S). The normalized sucrose gradient profile for fully associated conditions (8 mM Mg^2+^) was subtracted from each normalized profile to exclude 70S ribosomes in the peak for 50S. For each obtained curve, the area sums were calculated for peaks corresponding to 50S and 30S subunits, and sums were normalized to their maximum value. The obtained values were plotted on a graph.

The interface areas were calculated using the PISA program [24].

The contact amino acids and distance between uL14 and RsfS were obtained using Protein Interactions Calculator [25].

The solvent-accessible surfaces were calculated using the CASTp service [22].

The figures were prepared using Chimera [26] and PyMOL [27].

## 5. Conclusions

We conclude that the contact around Y98 of RsfS and T96 of uL14 from *S. aureus* is critical for RsfS’s anti-association activity and its binding to the 50S subunit. We confirmed this experimentally using sucrose gradient profiles and a cell-free in vitro translation assay. The results we found correlate well with previously published data.

## Figures and Tables

**Figure 1 ijms-23-10931-f001:**
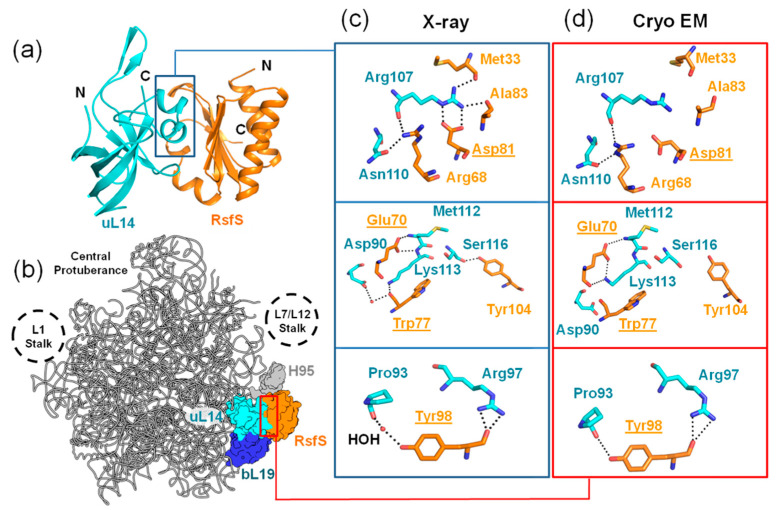
The interaction of RsfS with uL14 and the 50S ribosomal subunit from *S. aureus*. (**a**) The crystal structure of the RsfS–uL14 heterodimer (PDB: 6SJ5). (**b**) The cryo-EM structure of the 50S–RsfS complex from *S. aureus* (PDB: 6SJ5). (**c**,**d**) The contacting amino acid residues at the interface between uL14 and RsfS from X-ray and cryo-EM structures, respectively. Potential H-bonds are represented by dashed lines. Water molecules (HOH) are represented by red spheres. The chosen amino acid residues for replacement are underlined.

**Figure 2 ijms-23-10931-f002:**
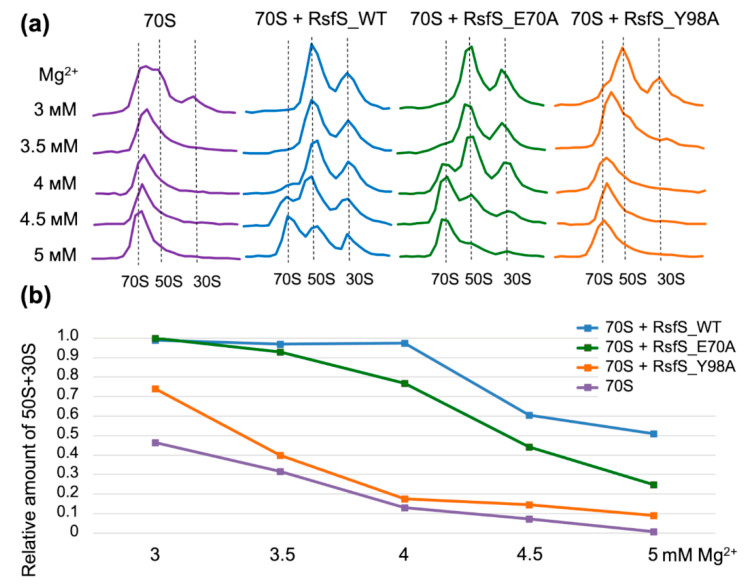
The influence of RsfS amino acid substitutions on ribosomal subunit dissociation at different concentrations of Mg^2+^. (**a**) Sucrose gradient profiles of the vacant 70S ribosomes with the wild-type RsfS (WT) and its mutants at different concentrations of Mg^2+^. (**b**) The relative amount of dissociated ribosomal 50S + 30S subunits (peak squares under the sucrose gradient profile) to the full number of subunits (50S + 30S + 70S) depending on the magnesium concentration.

**Figure 3 ijms-23-10931-f003:**
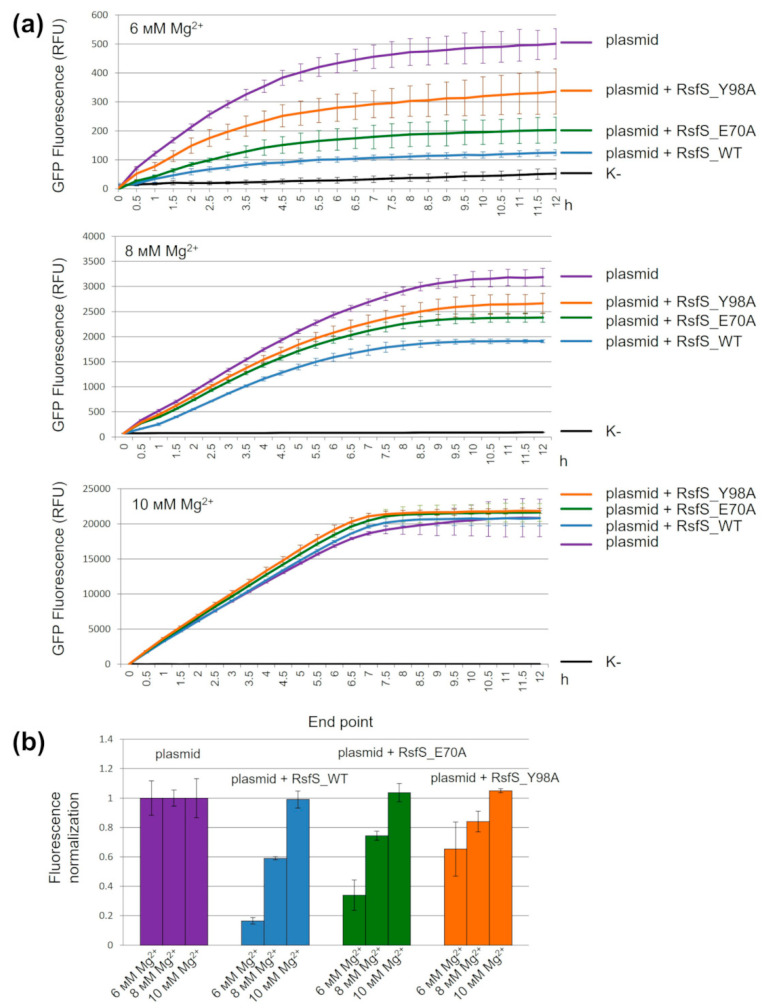
The influence of RsfS amino acid replacements on GFP cell-free translation with a variable Mg^2+^ concentration. (**a**) Kinetics experiments of GFP fluorescence in a cell-free in vitro translation assay. (**b**) Endpoint GFP fluorescence. «K-», cell-free system mix without the addition of a reporter plasmid (black); «plasmid», cell-free system mix with the addition of a reporter plasmid (violet); «plasmid + RsfS_WT», cell-free system mix with the addition of a reporter plasmid and the wild-type RsfS (blue); «plasmid + RsfS_E70A», cell-free system mix with the addition of a reporter plasmid and the mutant RsfS_E70A (green); «plasmid + RsfS_Y98A», cell-free system mix with the addition of a reporter plasmid and the mutant RsfS_Y98A (orange). Error bars indicate the standard deviation (n = 3).

**Figure 4 ijms-23-10931-f004:**
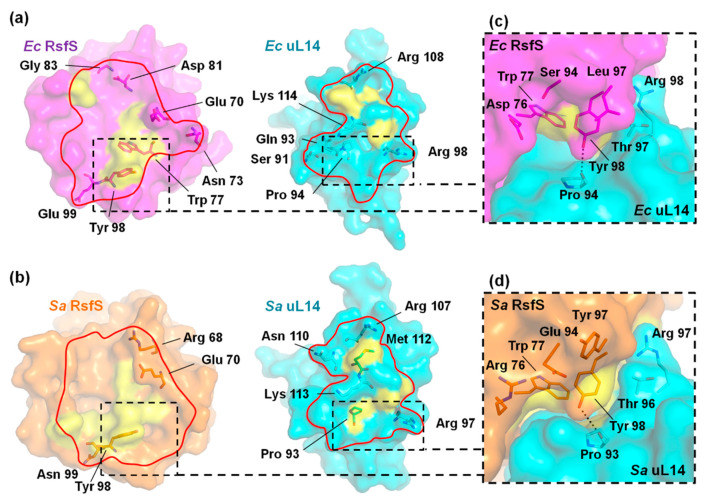
The interaction interfaces of the RsfS–uL14 heterodimer in the cryo-EM complex: (**a**) *E. coli* (PDB: 7BL4); (**b**) *S. aureus* (PDB: 6SJ6). The interaction surfaces between RsfS and uL14 are represented by the red border. Hydrophobic regions are shown as yellow surfaces and contacting H-bond-forming amino acids are shown as sticks. The heterodimer contacting surfaces around Y98 of RsfS from *E. coli* (**c**) and *S. aureus* (**d**) are shown in the zoomed boxes. H-bonds are represented by black dashed lines; the water molecule is represented by a red ball.

## Data Availability

Not applicable.

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
