# Peer review of "Y98 Mutation Leads to the Loss of RsfS Anti-Association Activity in Staphylococcus aureus"

_ijms, 2022, doi:10.3390/ijms231810931_

Round 1

Reviewer 1 Report

Fatkhullin and colleagues report that a point mutation (Y98) in the ribosomal silencing factor RsfS, located within the interface that RsfS forms with its binding partner uL14 on the large ribosomal (50S) subunit, prevents the well-established antiassociative effect of RsfS.

Major points:
1. The authors describe the biochemical effect of RsfS as “antiassociative”. However their experiments (Fig. 2) rather demonstrate a dissociative (70S dissociation) effect, which however, could be interpreted as prevention of re-association. For clarity, they should investigate how RsfS and the mutants E70A and Y98A inhibit association of the subunits (70S formation), when first added to isolated 50S, which is then mixed with purified 30S. Subsequent sucrose gradient runs should reveal the antiassociative effect and exhibit quantifiable differences.

2. Does overproduction of the RsfS mutants (E70A and Y98A) affect the growth of S. aureus, when compared with overproduction of wt RsfS? Spot tests and growth in liquid media are recommemded.

Minor points:

1. Is RsfS really essential?

2. The abbreviation RS for vacant 70S ribosomes appears irritating, and should be replaced by any other meaningful term.

3. Would a small molecule that is designed according to the authors strategy, attack specifically the interaction of S. aureus´s RsfS with the 50S subunit? Would it be expected to be a species-specific drug?  

4. The headline should mention the mutated residue Y98.

5. Language should be checked carefully.

6. Line 243: replace unconservative by non conserved.

Reviewer 2 Report

The amnuscript by Bulat Fatkhullin and co-authors describes an original experimental study of the RsfS protein. This protein binds ti the ribosomal protein L14 to serve as anti-associative factor. On the basis of structural data authors selected several residues of S.aureus RsfS for mutagenesis. Mutant forms that appeared to be soluble were cheched for anti-associative activity and translation inhibition activity at a range of magnesium ions concentrations. As a result residue Y98 was found to be important for the activity of the factor.

Results are new and scientifically sound. For increase in the clarity of the presentation, I would suggest to include sequence alignments of RsfS and L14 and discuss the conservation of contacts/mutated residues described in the study. 

Additional minor comments:

line 97 magnesium concentrations, not amounts

line 176 not "random", but "single" 

Round 2

Reviewer 1 Report

The authors adequately addressed all criticisms.